# Predicting the structure of large protein complexes using AlphaFold and Monte Carlo tree search

Patrick Bryant [1,2] ✉, Gabriele Pozzati [1,2], Wensi Zhu[1,2], Aditi Shenoy [1,2], Petras Kundrotas[1,3] & Arne Elofsson [1,2]

AlphaFold can predict the structure of single- and multiple-chain proteins with very high accuracy. However, the accuracy decreases with the number of chains, and the available GPU memory limits the size of protein complexes which can be predicted. Here we show that one can predict the structure of large complexes starting from predictions of subcomponents. We assemble 91 out of 175 complexes with 10–30 chains from predicted subcomponents using Monte Carlo tree search, with a median TM-score of 0.51. There are 30 highly accurate complexes (TM-score ≥0.8, 33% of complete assemblies). We create a scoring function, mpDockQ, that can distinguish if assemblies are complete and predict their accuracy. We find that complexes containing symmetry are accurately assembled, while asymmetrical complexes remain challenging. The method is freely available and accesible as a Colab notebook https://colab.research.google.com/github/patrickbryant1/MoLPC/blob/master/MoLPC.ipynb.

Large protein complexes govern many cellular processes, performing complicated tasks such as mRNA splicing[1], protein degradation[2] or assisting protein folding[3]. By incorporating protein-interaction information from many co-purification experiments, the human protein complex map, hu.MAP 2.0[4], provides a set of 4779 complexes with more than two chains. However, only 83 of these complexes are present in PDB. There are only 372 structurally resolved human protein complexes with over two chains, and of the 3130 eukaryotic core complexes in CORUM[5] only 800 have homologous structures covering all chains in PDB, suggesting a gap in our structural knowledge of protein complexes.

There are only 265 hetero and homomeric, non-redundant complexes in the PDB with 10–30 chains. Although it is unknown how many large complexes may exist, following the relationship between the known human complexes from hu.MAP and the structural coverage of these, one can extrapolate that there may indeed be a low structural coverage across different species.

There are at least three approaches[6] for modelling the structure of protein complexes, template-based modelling[7], shape complementarity docking[8] and integrative modelling[9,10]. Template-based modelling and docking methods have recently been shown to be outperformed by a combined fold and docking methodology using AlphaFold[11] for dimeric complexes, even if the bound form of each monomer is known[12]. Further, few docking programmes handle more than two protein chains, i.e., these methods are not suitable for building large complexes with no close homology to known complexes. There is currently (to our knowledge) no available docking benchmark for complexes with more than two chains, and previous studies only report results on a few examples[13,14].

Assembling large protein complexes with integrative modelling generally requires electron density maps or other experimental information to guide the assembly process[9,15]. This type of guided assembly is typically based on a Markov process[9] or Gaussian mixture models[16], where many different potential configurations are explored and scored. This process makes it possible to assemble complexes with up to 1000 protein chains[17]. However, obtaining electron density maps can be very difficult, as some protein complexes are hard to express, purify and crystallise. Still, many recent assemblies of large protein complexes exist, such as the human nuclear pore complex[18] and 26 S proteasome[19].

[1]Science for Life Laboratory, 172 21 Solna, Sweden. [2]Department of Biochemistry and Biophysics, Stockholm University, 106 91 Stockholm, Sweden. [3]Center for Computational Biology, The University of Kansas, Lawrence, KS 66047, USA. ✉e-mail: patrick.bryant@scilifelab.se

The only deep learning method primarily designed to predict the structure of more than two protein chains is AlphaFold-multimer[20]. This method has been trained on proteins of up to nine chains or 1536 residues and can predict complexes of up to a few thousand residues, where memory limitations come into play. However, the performance declines rapidly for proteins with over two chains (Supplementary Fig. 4). Predicting the structure of larger complexes is thereby currently not feasible. An alternative approach could be to predict the structure of subcomponents of large complexes and then assemble them. We have earlier shown that it is possible to manually assemble large complexes from dimers in a few cases[21].

In vivo, all components of large protein complexes do not assemble simultaneously, but stepwise[22], due to the presence of homologous protein chains and potential interfaces that need to be buried before subsequent chains can be added.

Here, we explore the limitations of AlphaFold for predicting protein complexes with 10–30 chains and create a graph-traversal algorithm that excludes overlapping interactions, making it possible to assemble large protein complexes in a stepwise fashion.

## Results

Here, we begin with an outline of the protein complex assembly using Monte Carlo tree search (MCTS). We then explore the success rates using either AlphaFold-multimer version 2[20] (AFM) or the FoldDock protocol[12] using AlphaFold[11] (AF). First, we examine the use of predicted subcomponents of the native dimeric or trimeric subcomponents. We then continue without assuming knowledge of the interactions, presenting the final protocol based on all possible trimeric subcomponents.

### Complex assembly

To analyse the possibility of assembling large protein complexes, we extracted all high-resolution non-redundant complexes from the PDB with more than nine chains not containing nucleic acids or interactions from different organisms (175 in total). We start by analysing the possibility of assembling these protein complexes assuming that exact pairs of interactions between protein chains are known. Using either AFM or FoldDock, we predict the structure of all unique pairs of interacting protein chains as subcomponents and create assembly paths from these.

As an example, the assembly of 6ESQ (acetoacetyl-CoA thiolase/HMG-CoA synthase complex) is shown in Fig. 1, using subcomponents predicted with AFM. The process starts from the two dimers, AC and CH, creating the trimer ACH through superposition using the chain C present in both dimers. Next, chain L is added through a connection with H (superposition using chain H); after that, chain J through a connection with L; this process then continues until the entire complex is assembled according to the outlined path.

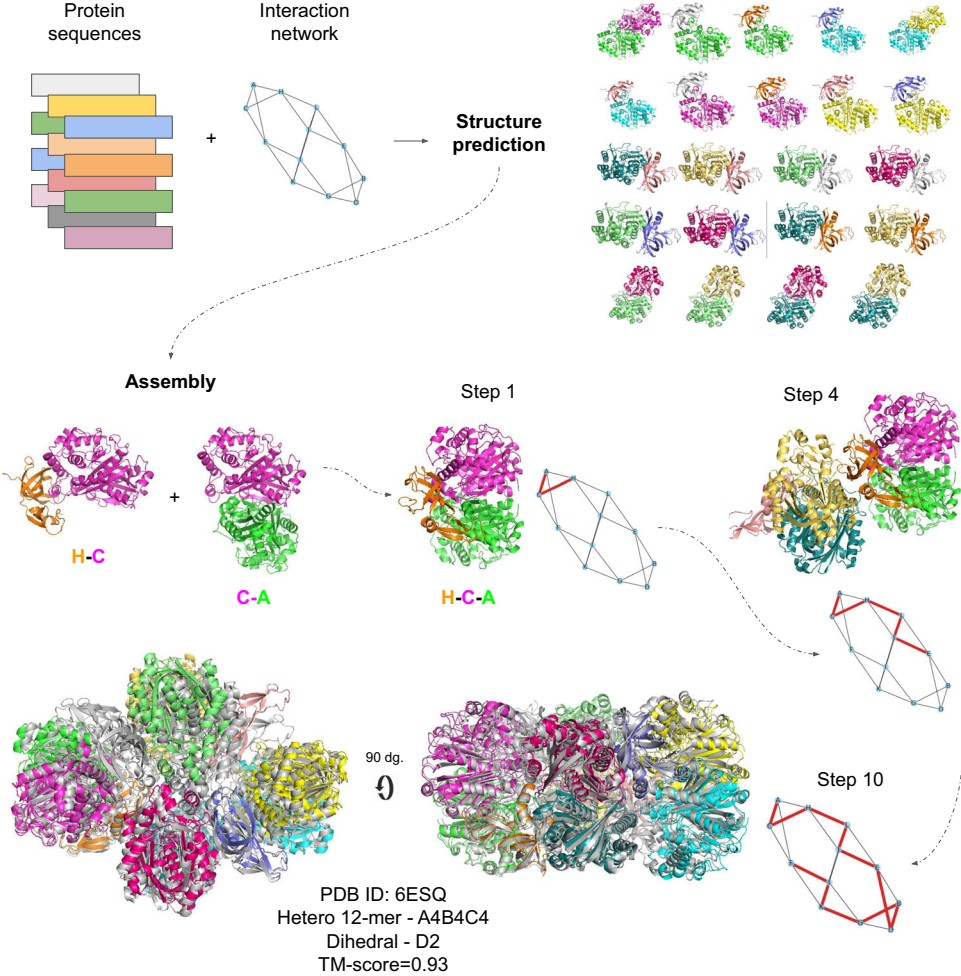

**Fig. 1 | Assembly principle for the acetoacetyl-CoA thiolase/HMG-CoA synthase complex (complex 6ESQ).** The structure of all interacting chains is predicted by protein sequences from each chain and the interaction network. From these predictions, an assembly path is constructed using the predictions as a guide. In each step, one new chain is added through a network edge resulting in a sequential construction of the complex. The taken path is outlined in red. The complete assembly is shown in overlap with the native complex (grey). The resulting TM-score is 0.93 using subcomponents from AFM (shown) and 0.92 using FoldDock (not shown).

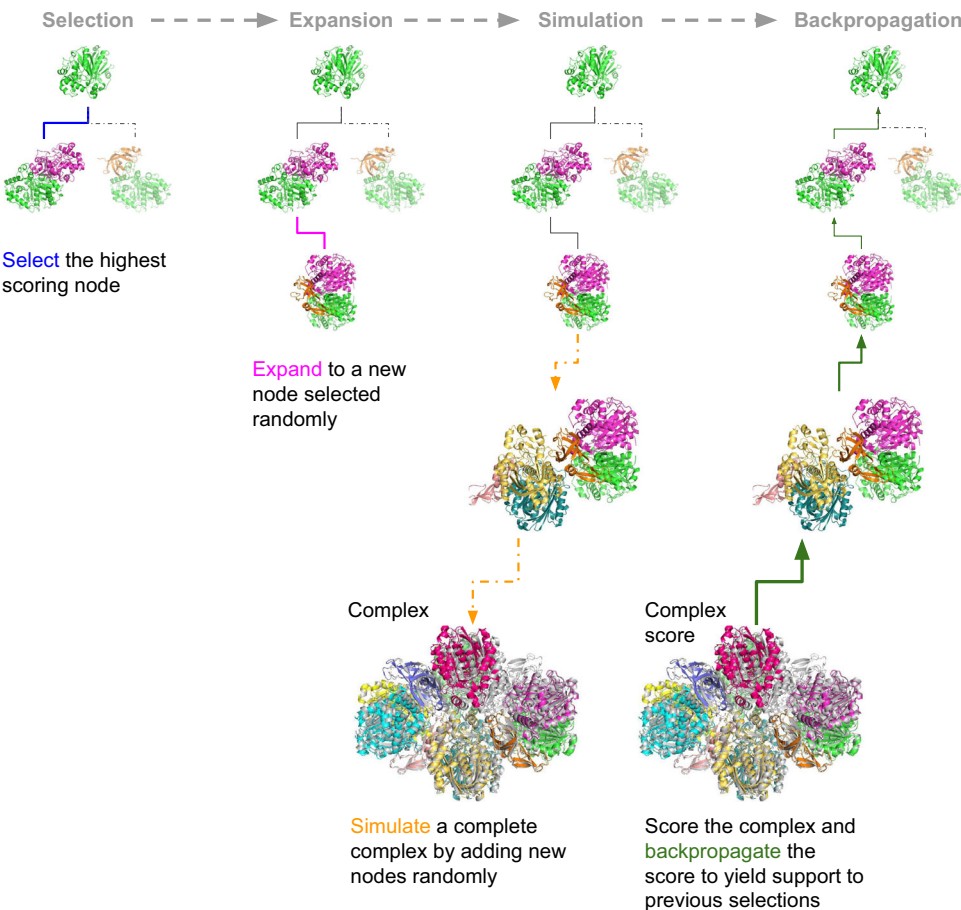

**Selection** – – – ➤ **Expansion** – – – ➤ **Simulation** – – – ➤ **Backpropagation**

Select the highest scoring node

Expand to a new node selected randomly

Complex

Complex score

Simulate a complete complex by adding new nodes randomly

Score the complex and backpropagate the score to yield support to previous selections

**Fig. 2 | Monte Carlo tree search.** Starting from a node (subcomplex) a new node is selected based on the previously backpropagated scores. From this node, a random node is added (expansion). A complete assembly process is then simulated by adding nodes randomly until an entire complex is assembled or a stop caused by too much overlap is reached. The complex is scored, and the score is backpropagated to all previous nodes, which yields support for the previous selections. The final result is that the nodes most likely to result in high-scoring complexes are joined in a path containing all chains. The principle for the complex 6ESQ is shown.

During the first part of this paper, we assume that the interaction graph is known, i.e., we limit the assembly paths only to include interactions existing in the complex. Although this is a simplification, the number of assembly paths is still huge, and it could at least be theoretically possible to obtain this information from other types of experiments[23] or predictions[4]. Next, potential assembly paths are created by starting at a randomly selected chain and adding all possible connections through superposition. Often overlaps occur among the predictions due to imperfect subcomponents, resulting in that, e.g. atoms from chains B and C occupying the same spatial position in a given complex ABCD. Therefore, an assembly path is discontinued when over half of the alpha carbons from two different chains are within 5 Å of each other. An assembly path is complete when all chains in a complex can be linked together. For 6ESQ, the assembly results in a model with a TM-score of 0.96.

**Monte Carlo tree search**

Due to the high number of possible paths to explore, searching all paths is unfeasible. Therefore, we search for an optimal path using MCTS[24] (Fig. 2), which has been applied successfully to solve a variety of game-related problems[25,26]. Starting from a randomly selected chain (node) chains are added randomly to expand the path, thereby creating new nodes. From these expansions, complete assemblies are simulated. Simulations are stopped when no additional subunits can be added, see Methods. The simulated assemblies are scored by their cumulative mpDockQ (multiple-interface predicted DockQ; average interface plDDT times the logarithm of the number of interface contacts, Methods section) score, and the scores are backpropagated to yield support for the previous selections. The path with the most support is selected, creating a complex that is the most likely to be correct. Due to the statistical nature of the search procedure, no aspect of a specific complex is being "learned" in the backpropagation, i.e., all 175 complexes can be used for evaluation.

**AFM vs. AF using pairwise interactions**

Only fifteen out of 175 complexes could be assembled to completion using native pairwise interactions with both AFM and FoldDock (Supplementary Fig. 1a). The assemblies based on FoldDock and AFM are partly complementary, meaning that the subcomponents of either AFM or FD can be used to assemble a complex. The results suggest that if a complete path can be found the models from AFM are slightly better (median TM-score = 0.83) compared to the FoldDock models (median = 0.77).

The AlphaFold-multimer version 1 (AFM-v1) modelling pipeline often caused clashes (Supplementary Fig. 1b), resulting in atoms from different chains occupying the same positions, which is why a new version was developed (version 2). Defining clashes as atoms from different protein chains being within 1 Ångström from each other, 26.7% (175/656) of the AFM dimers contain clashes and 6.3% (41/656) for the FoldDock pipeline. Even though there are more clashes in the AFM predictions, the final assemblies turn out to be of better quality, suggesting that the subcomponents are accurately predicted, for details see below.

## Limited conformational sampling in dimers

During assembly, the additive relative orientation of different protein chains can result in overlaps, due to predictions not being entirely correct. One cause of overlaps during the assembly process is due to that not all conformations of chain interactions are correctly predicted, resulting in wrong or missing interfaces in some dimers. As an example, we investigated 1A8R, a homo 10-mer. When predicting unique pairwise interactions, only one type of dimeric conformations can be found, but in the complex, each chain has at least two different types of interactions with other chains. This means that it is impossible to assemble the entire complex from the predicted dimers. The overlapping interfaces can, here, be circumvented by predicting trimeric interactions, thereby generating alternative interfaces (Supplementary Fig. 1c). We, therefore, continue with predicting trimeric subcomponents and use all three dimers from a trimer for assembly.

## Complex assembly using trimeric interactions

Using the FoldDock protocol and AFM all native trimeric interactions were predicted for all complexes. Using the unique native trimers, the clashes are more frequent than using the dimers, 37.1% (829/2234) for AFM, while for the FoldDock pipeline, the corresponding fraction is 23.2% (520/2242). All dimeric interactions were extracted from the trimers and assembly paths were constructed as previously. Out of 175 complexes, 58 (33%) and 55 (31%) could be assembled to completion with median TM-scores of 0.80 and 0.74 for FoldDock and AFM, respectively (Fig. 3a, Supplementary Table 1). Compared with the guided dimer TM-scores the guided trimer approach results in 46 additional complexes and a higher median TM-score for FoldDock, while three could not be assembled. For AFM, 43 additional complexes are produced with the guided trimer approach, while three are missed, but the median TM-score is lower than for dimers.

In many cases, the exact interactions of all protein chains are not known, only that a set of chains interact[4]. After applying MCTS to protein complexes where we have assumed knowledge of interactions, we now turn to the more challenging (and realistic) problem of predicting the complexes without knowing interactions (full approach). In addition to the problem of possibly incorrectly identified interacting pairs, this also increases the number of possible erroneous paths dramatically. Anyhow, we find that 91/175 (52%) of structures can be assembled with a median TM-score of 0.51 (Fig. 3a) using all possible trimeric interactions with FoldDock, while for AFM, only 74 complexes are complete with a median TM-score of 0.61.

When both trimer approaches have complete assemblies with FoldDock ($n = 53$), the median scores are 0.76 and 0.80 for the full and native trimer approaches, respectively. For AFM, the corresponding scores are 0.77 and 0.79 (45 complexes). Using native interactions thereby results in slightly higher scores overall, but to a lower fraction of complete assemblies. Models generated by FoldDock outperforms AFM overall, mainly because of fewer complete assemblies from AFM (Fig. 3a, Supplementary Table 2). We also included a comparison with Multi-LZerD[27] and Haddock[28], providing the real chain structures as input. For Haddock, 77 complexes were completed with a median TM-score of 0.29 (Supplementary Fig. 2, notably no model has TM-score ≥0.5). Unfortunately, for Multi-LZerD, we were unable to complete the docking for any complex in our dataset (Methods).

To analyse the possibility to distinguish when a complex is assembled to completion and has a high TM-score (≥0.8, $n = 30$), we analyse the ROC curve (Fig. 3b) as a function of; the average interface plDDT (predicted lDDT from AF), the number interface residues, contacts and interactions between chains normalised with the number of chains in each complex, and the average interface plDDT times the logarithm of the number of interface contacts. The plDDT·log(contacts) results in the highest AUC value (0.77). We fit a sigmoidal curve using the plDDT·log(contacts) and the TM-score, creating the mpDockQ score (multiple-interface predicted DockQ, see Methods

section). When the mpDockQ tends to be high, so does the TM-score and completeness of the complex (Fig. 3c). This suggests that mpDockQ can be used to select when a complex is complete and how accurate it is.

Figure 3d shows the TM-score distribution of assembled complexes using all possible trimeric subcomponents from FoldDock and examples at different TM-score thresholds. The type of symmetry of the complexes and the accuracy of the subcomponents strongly impact the outcome (see below). We find that complexes with any type of symmetry can be assembled with high accuracy, displaying the applicability of MCTS for symmetrical complexes (Supplementary Fig. 3). Obtaining complete complexes with very high TM-scores (≥0.8) is critical, as large complexes that are not entirely correct are not likely to provide biologically meaningful insights. It is also possible that some complexes with low TM-scores are accurate assemblies, but in a different conformation (or biological unit) than found in the PDB (e.g. 5TRM has octahedral symmetry, but is assembled in a dihedral configuration).

## Aspects affecting the assembly

To answer why some complexes can be assembled with high accuracy and others not, we analyse the kingdom, the number of total chains, the oligomeric type (hetero or homomer), the number of effective sequences (Neff), the subcomponent accuracy for each complex, the type of symmetry and the interface accuracy between predicted and assembled interfaces (Fig. 4). We performed this analysis for the complexes assembled with predicted native trimers due to the high redundancy of subcomponents in the blind approach. We use the MSAs and resulting complexes from FoldDock, as it is impossible to obtain the paired MSAs and features thereof in the AFM pipeline (the feature representation is created and processed as structures are predicted).

Bacteria is the most abundant kingdom and displays the highest fraction of complete assemblies (29/85) with a median TM-score of 0.85 (Fig. 4a). Eukaryota, Viruses and Archaea have 17/63, 8/12 and 4/15 with median TM-scores of 0.75, 0.44 and 0.92, respectively. Most complete assemblies have fewer chains and are of homomeric type (Fig. 5b, c), although the spread in TM-score is large. The TM-scores are higher for the complexes with higher (over 500) average Neff values, which corresponds well with findings for heterodimeric complexes[12] (Fig. 4d). When analysing how far toward completion the assemblies go, one finds that most complexes are 90–100% complete (Fig. 4e). There appears to be a weak decreasing trend in TM-score with completion suggesting that smaller subcomplexes may be accurate, although the complete complex cannot be assembled. The average TM-score of the subcomponents (Fig. 4f) provides the most evident explanation of when an assembled complex is accurate. When the subcomponents display high accuracy, so does the assembled complex. This is true for both complete and incomplete assemblies and highly accurate complexes (TM-score ≥0.8) can be selected with AUCs of 0.88 and 0.85, respectively (Supplementary Fig. 5).

The symmetry of the complexes is also found to be significant (Fig. 4g). The dominating symmetry (Dihedral) is also the one with the highest number of complete assemblies (27/70 complete, median TM-score = 0.80), while very few asymmetric complexes (2/26) are assembled to completion. These have low TM-scores (median = 0.49), suggesting that only symmetrical complexes can be assembled successfully using subcomponents and MCTS (median TM-score for all complexes with any symmetry = 0.80, $n = 58$). The asymmetric complex displayed in Fig. 4g (1L0L, https://www.rcsb.org/structure/1l0l) has a TM-score of 0.75, however, and most chains are in their correct positions with the exception of one membrane helix and a small cytosolic chain causing the decrease in TM-score.

The only Octahedral complete assembly has a TM-score of 0.99 and the Tetrahedral median is 0.92. The two complexes with helical

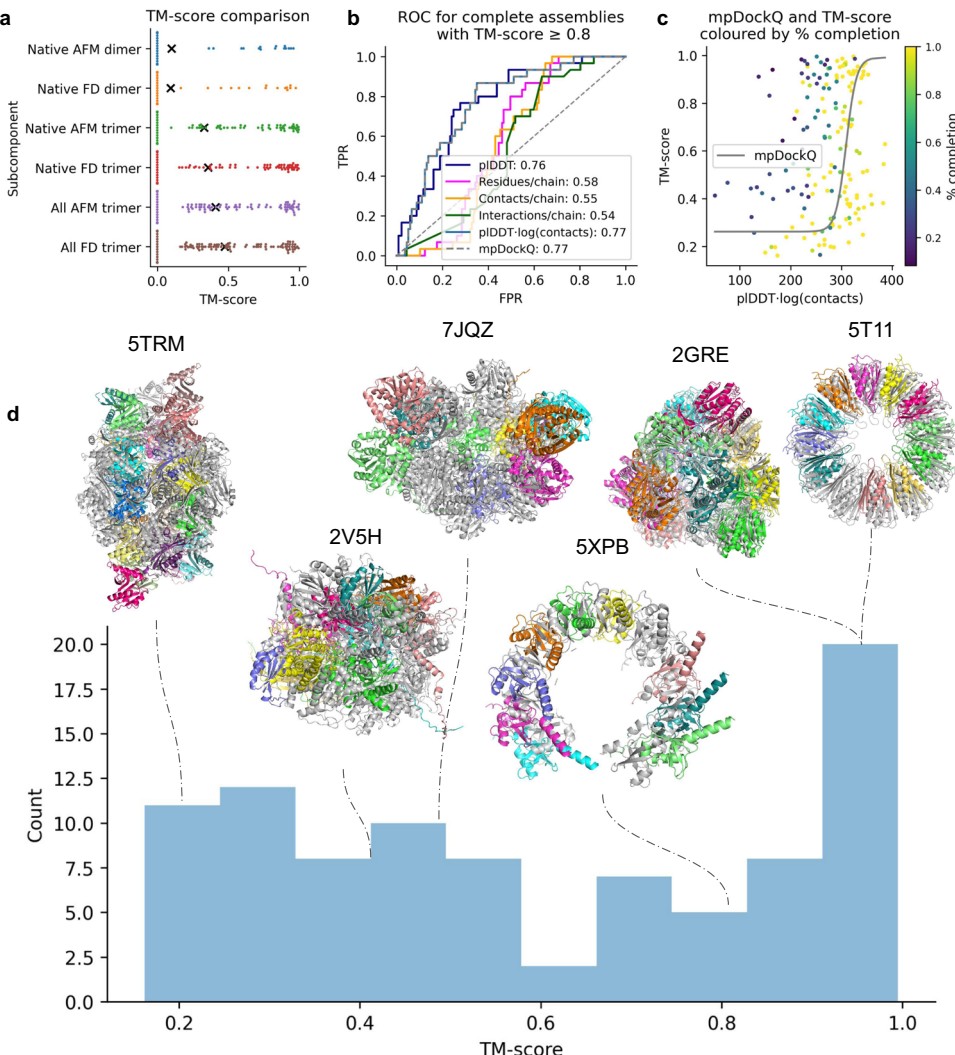

**Fig. 3 | Analysis of assembly success using different methods. a** TM-scores for the complexes that could be assembled to completion using FoldDock (FD) or AFM and predicted native dimeric, native trimeric and all trimeric subcomponents, respectively. The complete set of complexes from the three approaches ($n = 108$) is shown, with scores of zero representing missing complexes for each approach. The points display the TM-score of the individual complexes and the black "x" marks the average scores. The average TM-scores is 0.09 vs 0.10, 0.36 vs 0.33 and 0.47 vs 0.41 using native dimers, trimers and all trimers for FoldDock vs AFM, respectively. FoldDock thereby outperforms AFM overall. The median scores are low due to the missing complexes between the approaches. Considering only the successful assemblies using native dimers, trimers and all trimers, the median scores are 0.77, 0.80 and 0.51, respectively. **b** Complex scoring using all trimers as subcomponents. ROC curve, where positives ($n = 30$) are complete assemblies of TM-score ≥0.8, as a function of the average interface plDDT, the number of interface residues and contacts normalised with the number of chains in each complex, the average

interface plDDT times the logarithm of the number of interface contacts and mpDockQ (see c). The best separators are plDDT·log(contacts) and mpDockQ, both with AUC 0.83. **c** TM-score vs the best separator in b), plDDT·log(contacts), coloured by the fraction of completion for the assemblies ($n = 175$). The solid grey line represents a sigmoidal fit creating the mpDockQ score (see Methods section). When the mpDockQ tends to be high, so does the TM-score and % completion of the complex. This suggests that mpDockQ can be used to select when a complex is complete and how accurate it is. **d** TM-score distribution of the complete complexes ($n = 91$) assembled using all FoldDock trimers and examples at different thresholds. The assembled complexes (coloured by chain) are in structural superposition with the native ones (grey). The PDB IDs for the complexes shown and their corresponding symmetries and TM-scores are 5TRM (Octahedral, 0.22), 2V5H (Dihedral, 0.45), 7JQZ (Dihedral, 0.51), 5XPB (Helical, 0.82), 2GRE (Tetrahedral, 0.97) and 5T11 (Cyclic, 0.98). At TM-score 0.8, the assembled complex is similar to the native one.

symmetry display varying quality (TM-scores of 0.37 and 0.88, respectively). All types of symmetry display successful examples (Fig. 4g), suggesting that MCTS can assemble subcomponents successfully as long as some type of symmetry exists in the complex. The TM-score distribution divided by symmetry for the all-trimer approach using FoldDock follows a similar relationship to the one found here using the guided trimer approach (Supplementary Fig. 3).

As the assembly proceeds, new interfaces are generated. To provide an estimate of the quality of the assembled interfaces we compare these to those predicted directly with FoldDock using the DockQ score[29]. Only the complexes that could be assembled to completion and that had interfaces resulting from the assembly were analysed

($n = 55$). The predicted interfaces are found to be of higher quality on average (DockQ = 0.34 vs 0.24 for predicted and assembled interfaces, respectively).

**Assembling complexes with 4–9 chains**

So far, we have only addressed the possibility to assemble complexes with 10–30 chains, outside of current computational limitations. However, it is possible that the assembly approach with MCTS can be used for smaller complexes as well. To analyse the possibility to improve the accuracy of AFM end-to-end (E2E) on smaller complexes we created a set of 278 complexes with 4–9 chains that have less than 30% sequence identity to the proteins in the AFM training set (see

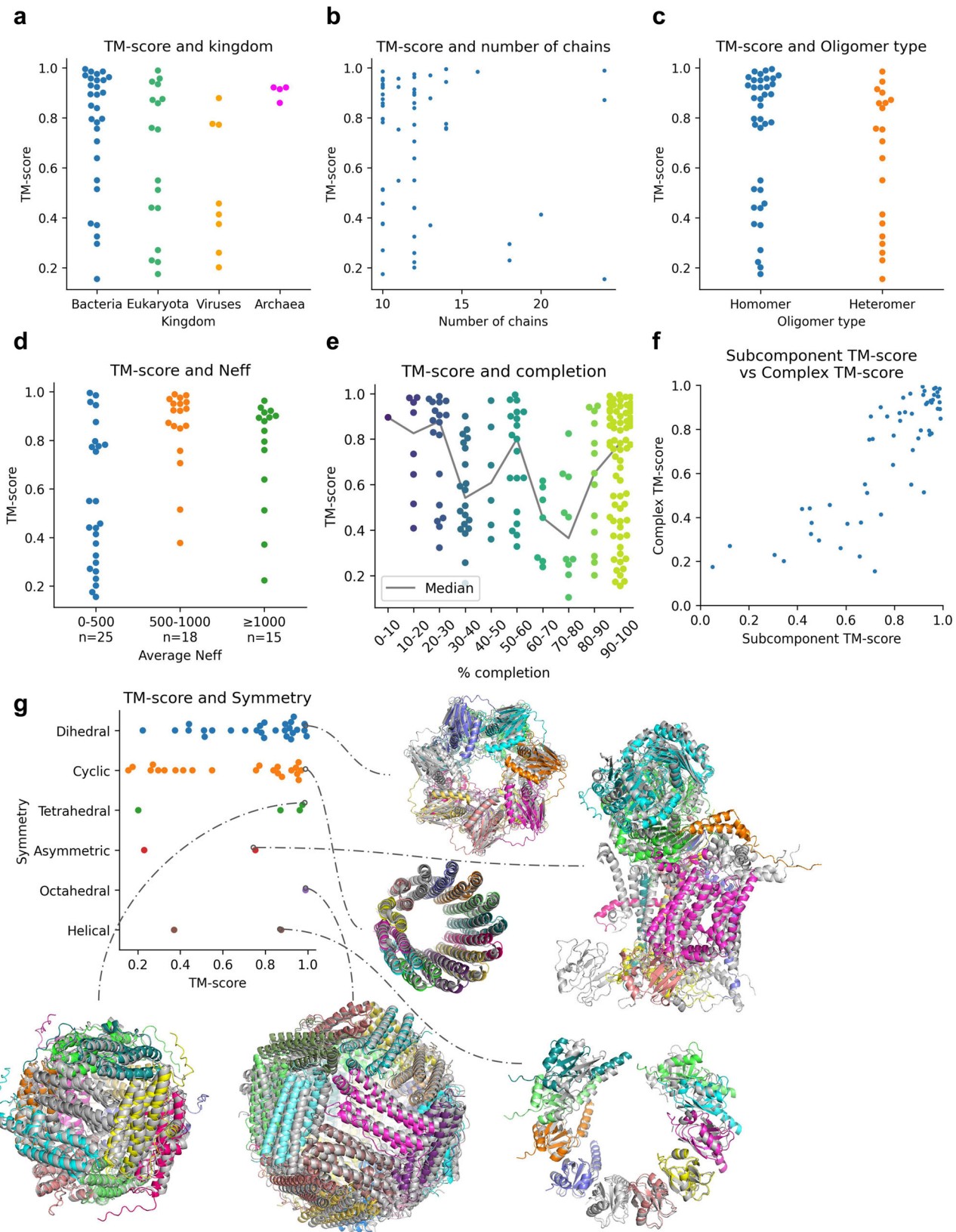

Methods). We predicted all possible trimeric subcomponents using FoldDock and assembled them using MCTS. We find that the performance of AFM E2E is higher than the MCTS assembly across all oligomers of 4–9 chains (Fig. 5). The performance is quite consistent for 4–7 chains, but drops for 8–9 chains (averages of 0.47 vs 0.58, 0.50 vs

0.54, 0.46 vs 0.56, 0.49 vs 0.67, 0.39 vs 0.41 and 0.35 vs 0.46 for AFM E2E vs MCTS 4–9 chains, respectively). The performance is low on average, suggesting there is much room for improvement and that predicting the structure of even small protein complexes is a problem that is not yet solved.

**Fig. 4 | Analysis of assembly characteristics using native trimers predicted with FoldDock. a** TM-score per kingdom for the complete assemblies ($n = 58$). Bacteria is the kingdom with the highest number of complete assemblies ($n = 29$) and reports a median TM-score of 0.85. Eukaryota ($n = 17$), Viruses ($n = 8$) and Archaea ($n = 4$) have median TM-scores of 0.75, 0.44 and 0.92, respectively. **b** TM-score vs. the number of chains for the complete assemblies ($n = 58$). **c** TM-score vs oligomer type, homomer ($n = 38$ out of 114) or heteromer ($n = 20$ out of 61), using complete assemblies. The homomeric complexes have a median TM-score of 0.86 and the heteromeric 0.73. **d** TM-score and Neff. Average TM-scores are higher for the complexes with over 500 in average Neff value. **e** TM-score and completion for all complexes ($n = 175$). The coloured points represent the scores within bins of 10%, and the grey line shows the median for each bin. **f** Average TM-score of sub-components vs TM-score of the whole complex for the complete assemblies ($n = 58$). When the subcomponents display high accuracy, so does the assembled complex (SpearmanR = 0.80). **g** Distribution of TM-scores and examples of the best assemblies for each symmetry type. The assemblies are coloured by chain, and the true complexes are in structural superposition in grey. The structures shown for each symmetry and the corresponding TM-scores are: 5OVS (Dihedral, 0.99), 2X2V (Cyclic, 0.97), 1DPS (Tetrahedral, 0.98), 1L0L (Asymmetric, 0.75), 1MFR (Octahedral, 0.99) and 5XPB (Helical, 0.88).

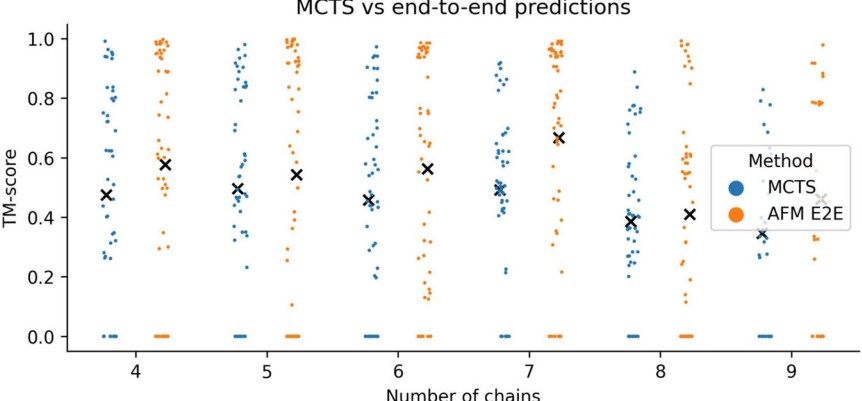

**Fig. 5 | Comparison of assembly with MCTS vs. AlphaFold-multimer on complexes with 4–9 chains.** Swarm plots displaying the TM-scores ($n = 278$, $n = 50$ for each oligomer except for the nonamers, which have $n = 28$) for assemblies using all possible trimers predicted with FoldDock (MCTS) and AFM end-to-end (E2E). Each point represents one complex with the mean TM-scores marked by a black "x". The points at zero for MCTS are those complexes that could not be assembled to completion ($n = 62$) and those for AFM E2E that were out of memory ($n = 67$). The averages are 0.47 vs 0.58, 0.50 vs 0.54, 0.46 vs 0.56, 0.49 vs 0.67, 0.39 vs 0.41 and 0.35 vs 0.46 for AFM E2E vs MCTS 4–9 chains, respectively.

## Discussion

To predict the structure of large complexes directly from sequence information is currently a difficult challenge. Here, we present a method that suggests that one possible approach is to predict subcomponents and assemble them into a larger complex. Using subcomponents predicted from native dimers, native trimers and all possible trimeric interactions, the median TM-scores are 0.77, 0.80 and 0.50, respectively (15, 58 and 91 complete out of 175 complexes). The scoring function mpDockQ can distinguish if assemblies are complete and predict their accuracy, making the blind approach with all trimers feasible. FoldDock based on AF outperforms AFM in predicting trimeric subcomponents in combination with being much faster (2–4 times, Methods), it can also be noted that AF was not trained for assembly of proteins either, yielding support to the robustness of this method.

We find that when the subcomponents are accurately predicted using native trimers, so are the complete assemblies. This suggests it is possible to assemble complexes as long as their subcomponents are accurate. The symmetry of the complex affects the outcome and some symmetries (Dihedral and Cyclic) are more abundant and easier to predict than others (Helical, Asymmetric). Not all trimers can be folded using two NVIDIA A100 Tensor Core GPUs with 40 Gb of RAM each. The limit of AF (and AFM) on this computational platform appears to be roughly 3000 residues, and 73/175 (42%) of all complexes are larger than that. We find that the assembly approach suggested here is outperformed by the current state-of-the-art AFM for complexes with 4–9 chains whenever it is possible to run AFM.

In summary, we have shown that assembling large complexes with different symmetries is possible using only protein sequence information and stoichiometry. Modelling large complexes in parts and assembling them converts the problem of predicting large complexes to the prediction of their subcomponents. This suggests an exciting future where models of all protein complexes in entire cells may be modelled.

One limitation of predicting protein complexes using the approach proposed here is stoichiometry. It is often not known how many copies of a protein are in a given complex, a requirement for the assembly. Once this limitation is overcome either by computational or experimental studies of complexes, it will be possible to assemble many different protein complexes, possibly in novel configurations.

## Methods

### Non-redundant complexes with 10–30 chains from the PDB

Since AlphaFold-multimer has a limit of nine chains or 1536 residues[20] in its training and testing data, and there is no available method validated for modelling larger complexes, we obtained all complexes with 10–30 chains from the PDB on 2022-01-10 to extend the current limit (Fig. 6a). First, we selected all complexes not containing nucleic acids with ≤3 Å (Ångström) resolution and with the experimental method X-ray crystallography or Electron Microscopy (1216). From these complexes, we require all chains to originate from the same organism (1027). We cluster all sequences from the complexes on 20% sequence identity using MMseqs2 (version edb8223d1ea07385ffe63d4f103af0eb12b2058e)[30] using this command:

```
MMseqs2 easy-cluster fastafile outname /tmp --min-seq-id
0.2 -c 0.8 --cov-mode 1
```

Using clustering, we ensure that no complex has all of its clusters overlapping with any other. We keep the complexes with the most clusters, resulting in the removal of subcomponents of larger clusters (265). E.g. if the sequences from complex 1 map to clusters A, B, and C and those of complex 2 map to clusters A, B, C,

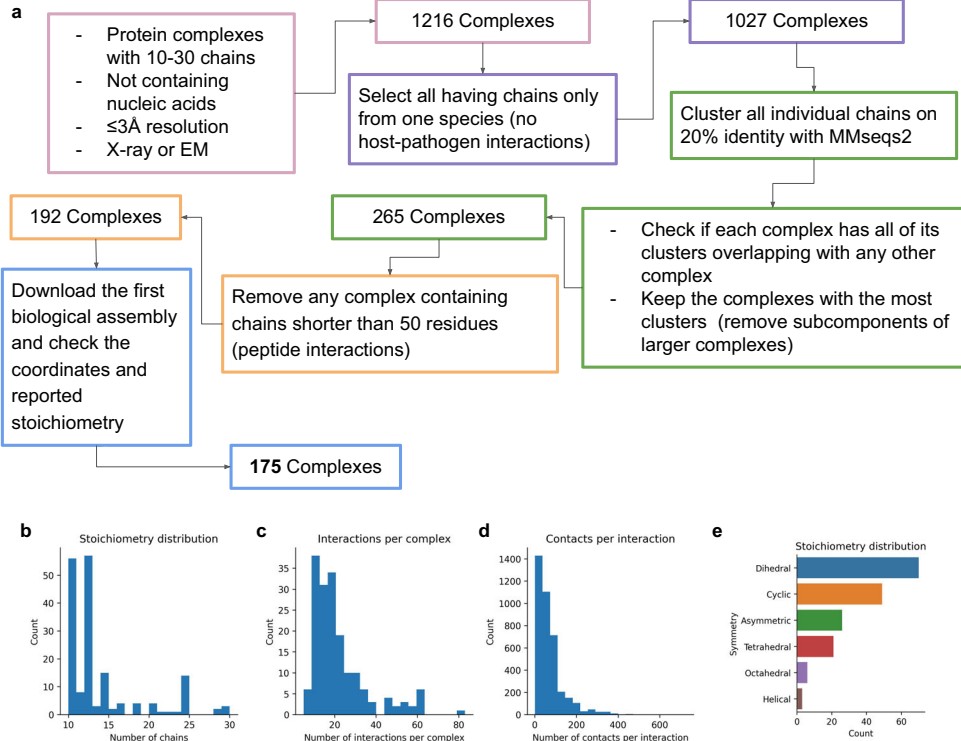

**Fig. 6 | Data selection process and statistics. a** Outline of the data selection process. **b** Distribution of the number of chains for the 175 complexes. Most complexes have 10–12 chains. **c** Distribution of the number of interactions between all chains in a complex (*n* = 175 complexes). On average, there are 22 interactions per complex. **d** Distribution of the number of contacts per interaction (*n* = 175 complexes). On average, there are 70 contacts per pair of interacting chains. **e** Distribution of the symmetry types of the complexes (*n* = 175 complexes). Dihedral complexes are the most common, followed by cyclic and asymmetric.

and D, then complex 2 will be kept and complex 1 excluded. After the clustering, we ensure that no complex contains any chain shorter than 50 residues (193 complexes), to remove protein-peptide interactions. We then download the first biological assembly[31] from each complex and check that the reported stoichiometry is correct and that the PDB files do not contain discontinuous chains, resulting in a total of 175 complexes. The distribution of the number of chains, the number of interactions between chains and the number of contacts per interaction can be seen in Fig. 6b–d, respectively. Most complexes have 10–12 chains, an average of 22 interactions and 70 contacts between each pair of interacting chains. The symmetry definitions were taken from the PDB annotation (global symmetry), Fig. 6e.

### Interaction network
To create interaction networks for the guided assembly of the complexes, interactions between different chains with CBs (CA for Glycine) within 8 Å from each other were extracted. Interactions are defined when 10% of the beta carbons (alpha carbon for glycine) of the shortest of two protein chains are within 8 Å from the other. On average, each interaction pair consists of 70 residue pairs, and within each complex, there are 22 interacting pairs of chains (Fig. 6c, d).

### Subcomponent and edge complexity
To assemble entire complexes, we predict all dimeric and trimeric interactions in a set of *n* chains.

The number of possible dimers follows:

$$D(n) = \frac{n!}{(n-2)!2!} = \frac{n(n-1)}{2} \tag{1}$$

The number of possible trimers follows:

$$T(n) = \frac{n!}{(n-3)!3!} = \frac{n(n-1)(n-2)}{6} \tag{2}$$

From these dimers and trimers, we extract all edges (pairwise interactions). The number of edges in D(*n*) dimers is D(*n*) and in T(*n*) trimers:

$$E(n) = \frac{T(n)(T(n)-1)}{2} \tag{3}$$

### Structural predictions of dimeric and trimeric subcomponents
AlphaFold-multimer (v2.0)[20] was run using all five models, with one structure per model, resulting in a total of five structures per prediction where the top-ranked one was selected for subsequent analyses. Four different MSAs are created by searching various databases with several genetic search programmes. Using jackhmmer from HMMER3[32], three different MSAs are created through searching the databases Uniref90 v.2020_01[33], Uniprot v.2021_04[34] and MGnify v.2018_12[35]. The fourth MSA is created by searching the Big Fantastic Database[36] (BFD from https://bfd. mmseqs.com/) and uniclust30_2018_08[37] jointly with HHBlits[38] (from hh-suite v.3.0-beta.3 version 14/07/2017). By using the species and genetic positional information, the results from the Uniprot search are paired. All results from the other searches are instead block-diagonalized. All of the created MSAs (one paired and three block-diagonalized) are used to predict the structure of a protein complex.

The FoldDock protocol[12], based on AlphaFold (v2.0)[11], was run as well. This protocol creates two MSAs constructed from a single search

with HHblits[38] version 3.1.0 against uniclust30_2018_08[37] using the options:

```
hhblits -E 0.001 -all -oa3m -n 2
```

The first of the two MSAs is constructed by extracting the organism identifiers (OX) from the resulting a3m file and pairing sequences using the top hit from each OX. The second is constructed by block diagonalizing the resulting a3m file. An extension to three chains was made here also, following the same pairing and block diagonalizing procedure as has been done for two chains. The folding was performed using AlphaFold model_1, 10 recycles and one ensemble structure. The recycles refer to how many times the intermediate output is fed back into the network and the MSAs are resampled. The ensemble structure entails how many times the information within the network is processed before it is averaged.

The structural prediction was performed on two NVIDIA A100 Tensor Core GPUs each with 40 Gb of RAM with a time limit of 24 h per prediction. Three sets of different subcomponents for the complexes with 10–30 chains were modelled, all native dimeric, all native trimeric and all possible trimeric subcomponents. For the complexes with 4–9 chains, all possible trimeric subcomponents were modelled to compare with predicting these directly using AFM E2E.

### Structural prediction limitations

For the complexes with 10–30 chains, the unique dimer subcomponents of 656/656 could be predicted for AFM and FoldDock, respectively. For FoldDock, 2242/2246 unique native trimers were predicted, and 2234/2246 for AFM. The four that did not work using FoldDock had the error message "Cannot create a tensor proto whose content is larger than 2GB." and the 13 that did not work for AFM ran out of time on two NVIDIA A100 Tensor Core GPUs each with 40 Gb of RAM with a time limit of 24 h per prediction. For the approach using all trimers, 8049/8049 unique subcomponents were successfully modelled using FoldDock. The all-trimer approach using AFM resulted in 7999/8049 unique subcomponents due to the 24 hour time limitation.

For the 278 selected complexes of 4–9 chains (see section "Non-redundant complexes from the PDB with 4–9 chains without homology to the AFM training set"), there were 2400 subcomponents for the all-trimer approach of which 2375 could be predicted with FoldDock. Memory limitations resulted in that 25 could not be predicted. In total, 62 complexes could not be assembled to completion using MCTS. For AFM E2E, 67 complexes could not be predicted due to memory limitations (8 Intel Xeon E5-2690v4 CPUs with a total of 82 Gb memory for the MSA generation and the same GPU limitations as above).

For the analysis of the entire dataset ($n = 1733$, supplementary Fig. 4) of non-redundant complexes with 2–9 chains predicted with AFM E2E, 207 failed (188 failed due to MSA memory limitations and 19 failed due to GPU memory limitations using the same hardware as above), resulting in 1526 successful oligomers in total.

### Path complexity

When considering all possible interactions in a complex, both dimeric and trimeric, one quickly realises that there are many possible paths that could connect all chains. Take the example of the maximum number of chains modelled here, 30. In the most extreme scenario, all of these are assumed to interact with each other. This means that starting at chain 1, it is possible to attach chains 2–30 (29 possibilities) and from these 28 possibilities for each node and so on.

If there are no overlapping interfaces in a complex of $n'$ nodes and $E(n)$ edges, the number of unique paths that contain all nodes follow:

$$P(n) = n'^{(n'-2)}, n' \geq 2 \tag{4}$$

Note that n′ here is the number of nodes extracted from the predicted subcomponents, which are more than the number of unique nodes since e.g. the trimers ABC and ABD both contain the nodes A and B. Equation 4 is exponential and thereby grows very fast. However, the overlaps will grow with the number of nodes as well, as it will be more likely to have overlapping interfaces with more edges.

According to Eq. 2, there are $\frac{30(30-1)(30-2)}{6} = 4060$ possible trimers for a complex of 30 chains. For each trimer, there are three possible edges, resulting in $4060 \cdot 3 = 12180$ edges in total. This means that the number of effective nodes is higher than the actual number of nodes. This is because e.g. chain A occurs many times in different trimers. E.g., ABC, ABD, ABE all have the possibility to have different interactions between A and B. Following Eq. 4 there will be $30^{28} \approx 2.3 \cdot 10^{41}$ possible paths at the upper bound considering all dimers from 30 protein chains (and many more considering all trimers). This is a very large number that is not possible to search in a feasible amount of time with our available computational resources. However, it is very unlikely this number of paths has to be explored due to overlaps in the subassemblies.

When the subpaths that contain overlaps are excluded during assembly, the number of possible paths reduces quickly. Let's assume there are only three possible interactions for each chain. Then the number of possible paths becomes much fewer, depending on how the network is connected. If all branches in a network contain unique chains (Fig. 7), there is in fact only one possible path that connects all chains. Still, there may be many possible paths to traverse to find this non-overlapping one that connects all chains. Therefore, we limit the number of paths searched at a given time point.

### Assembly procedure with Monte Carlo tree search

From the interactions in the predicted subcomponents, we add chains sequentially following a path through the interaction network (graph) constructed using MCTS[24]. MCTS applies a heuristic search method through a graph to find an optimal path (Fig. 8). MCTS consists of four different steps; selection, expansion, simulation, and backpropagation. It has been shown that sampling random paths to completion from a certain node (simulation) informs the best action at a certain position. To add new chains to a path, we use BioPython's SVD Superimposer[39]. As an example, if two pairwise interactions are A-B and B-C, we assemble the complex A-B-C by superposing chain B from A-B and B-C and rotating the missing chain C to its correct relative position. The MCTS procedure is outlined accordingly (for pseudo-code, see "Pseudocode for the Monte Carlo tree search algorithm" below):

1. Selection: start at a randomly chosen node 1 (e.g., chain A).
2. Expansion: obtain all edges $e^1, .., e^N$, deemed "children" to node 1 and create N different paths. Expand the new nodes added through the edges by randomly selecting new edges. If the new nodes do not have any edges, they are deemed "leaf nodes". In this case, the best scoring node according to Eq. 5 is selected and a new expansion is started from there. We expand all possibilities, ensuring convergence towards the best node selection at each position.
3. Simulation: add chains randomly to the path until the overlap criterium is obtained or the complex is complete. An overlap is defined as when over 50% of the alpha carbons in the shortest of two protein chains are within 5 Å from each other.
4. Backpropagation: score the simulated complex using Eq. 6. Update all "parent nodes" with this score. The simulation and

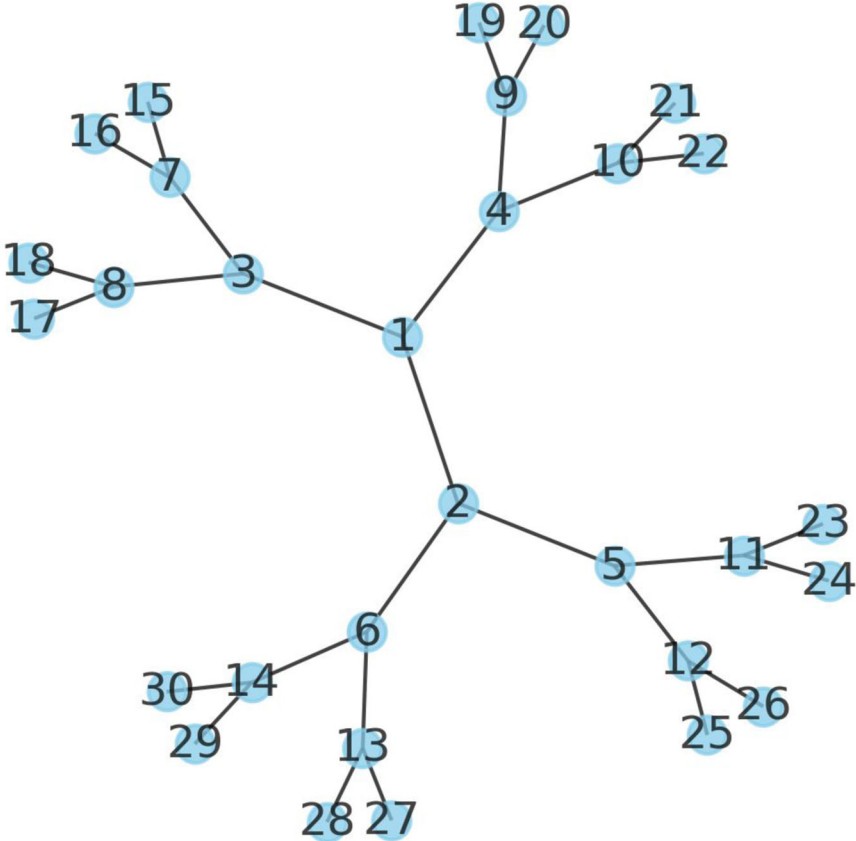

**Fig. 7 | Branch network of 30 chains all connected to two other chains.** There is only one path that connects all 30 chains (the network itself).

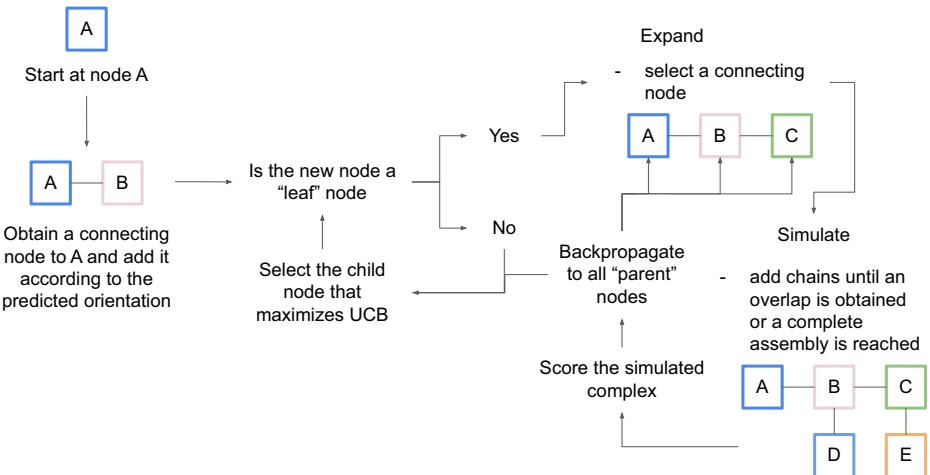

**Fig. 8 | Monte Carlo tree search (MCTS) procedure.** Starting at node A, a connecting node (chain) is selected and added according to its predicted orientation. If this node is a "leaf" node (a node that has not been expanded before), an expansion is performed. During the expansion, a new node is added and from this, an entire complex is simulated. The score from the simulation (Eq. 6) is backpropagated to all "parent" nodes of the expansion which is used to determine the UCB (Eq. 5) and thus select the best possible path.

backpropagation together provide an estimate of how well the parent node performs in terms of creating a successful assembly path.

The best child nodes are selected using the upper confidence bound (UCB) accordingly:

$$UCB = V_i + 2\sqrt{\frac{\ln N}{ni}} \qquad (5)$$

where $V_i$ is the average complex score (Eq. 6) of all nodes below node $i$, $N$ is the number of times the parent node has been visited and $n_i$ is the number of times the node being scored has been visited. The MCTS procedure is continued until all chains are complex or there are no more non-overlapping chains to add to the current path, after which the procedure is terminated.

The complex 6LNI could not be assembled using trimers due to no interactions between the chains being present in the

predictions. This protein is an amyloid protein and should thereby not occur naturally in the cell. For the assembly approach using all possible trimers, there are very many paths to assemble for some complexes. Six additional complexes (1MFR, 1S3Q, 3OJ5, 6J0B, 6NHT and 6PYT) could not be assembled due to time constraints (72 h on a 2.6 GHz processor).

## Pseudocode for the Monte Carlo tree search algorithm

```
Input:
    - Predicted edges and their structures
Output:
    - Largest possible assembled complex
    1 Select an edge connected to node (chain) A and make
    this the current node n
while(edge_available(n)):
    #Expand
    while(not is_leaf(n))
        n = best_child(n)
    end
    #Simulate and score using UCB
    score = roll_out(n)
    #Back propagate the score
    back_propagate(score)
end
#Return the path taken to reach the current node n
return get_path(n)
```

## Scoring

We score the interfaces of the complexes being assembled in the MCTS using:

$$\log_{10}(number\ of\ interface\ contacts) \cdot average\ interface\ plDDT, \quad (6)$$

as done when calculating the pDockQ score[12]. This score for multiple interfaces, we deem "multiple-interface predicted DockQ" or mpDockQ. The interface contacts are taken as beta carbons (alpha carbons for Glycine) from two different protein chains being within 8 Å of each other. These metrics are calculated for the entire interface of each chain, as in the DockQ[29] score for multiple interfaces. E.g., if chain A interacts with both chains B and C, the score is taken over both of these interfaces simultaneously. This is done for all interfaces and chains and summed over the entire complex. The complexes with the highest sums are favoured. Favouring complexes with higher scores, results in complexes with both larger interfaces and with more reliably predicted residues.

## Sigmoidal fit for mpDockQ

To create a continuous score for the multiple-interface DockQ (mpDockQ), we fit a simple sigmoidal function towards the TM-score (Fig. 3c) using the complete complexes assembled from trimeric sub-components and "curve_fit" from SciPy v.1.4.1[40] with the following sigmoidal equation:

$$mpDockQ = \frac{L}{1 + e^{-k(x-x_0)}} + b \quad (7)$$

where $x$ = average interface plDDT·log10(number of interface contacts) (Eq. 6) across all interfaces and we obtain $L = 0.728$, $x_0 = 309.375$, $k = 0.098$ and $b = 0.262$.

## Clashes

To analyse if the atoms from different chains in the same prediction overlap, we calculate the distance between all atoms in all chains in a given prediction. We count clashes as two-atom positions from different chains being within 1 Å of each other (the size of one hydrogen atom).

## MMalign

The DockQ[29] programme is too slow to be run on large complexes if all interfaces are to be compared (minutes-hours for a single complex). Therefore, the programme MMalign[41] is used to score entire complexes, as compared to the scoring of dimeric complexes with Fold-Dock previously[12]. MMalign performs optimal structural alignment between the model and native structures, computing a score (TM-score) normalised to be between zero and one, where one indicates a perfect match.

Since MMalign performs optimal structural superposition, it is also possible to evaluate models of different sizes. This is important since the predictions are based on full-length protein sequences (and to score incomplete assemblies), while the PDB structures generally do not contain all residues from these, meaning that loops and other disordered regions are not present in the PDB structures. This also means that for most proteins, the score can never be 1, depending on how similar the SEQRES sequence is to the sequence present in the PDB structure. Since we assess the real sequences here, our approach represents a more realistic modelling scenario.

## Number of effective sequences

The Neff is a measure of the information present in a multiple-sequence alignment. To calculate the Neff, we clustered sequences from each MSA independently (the paired versions) at 62% sequence identity, following the rationale behind the BLOSUM62 matrix[42]. The clustering was performed using MMseqs2 version fcf52600801a73e95fd74068e1bb1afb437d719d[30]rs was used to indicate the Neff. MMseqs2 was run with the following command:

```
MMseqs2 easy-cluster msa outname /tmp --min-seq-id 0.62 -c
0.8 --cov-mode 1
```

The clustering was done for all predicted subcomponents in each complex. To obtain a Neff score for each complex, we averaged the scores for all subcomponents.

## ROC curve

We create receiver operating characteristic (ROC) curves using the metrics average interface plDDT (predicted lDDT from AF), the number of interface residues, contacts, and interactions between chains normalised with the number of chains in each complex and the mpDockQ (multiple-interface predicted DockQ; average interface plDDT times the logarithm of the number of interface contacts). The positive examples are taken either as complete assemblies (when all native chains are present in an assembly) or above the median TM-score (only for mpDockQ). The metrics are used to distinguish between true and false positives (TP and FP, respectively) by creating thresholds of all possible metric values. From the thresholding we calculate the true- and false positive rates:

$$TPR = \frac{TP}{TP + FN} \quad (8)$$

$$FPR = \frac{FP}{FP + TN} \quad (9)$$

Using the thresholds and corresponding TPR and FPR, the TPR is plotted against the FPR. This creates a ROC curve. For each metric, the

area under the ROC curve (AUC) is computed as:

$$AUC = \int_{x=0}^{1} TPR\left(\frac{1}{FPR(x)}\right) dx \qquad (10)$$

## Non-redundant complexes from the PDB with 4–9 chains without homology to the AFM training set

The procedure for selecting complexes between 4–9 chains is different from that of selecting 10–30 chains due to homology reduction towards the AFM training set (trained on 2–9 chains in PDB before 2018-04-30[20]) and the resulting scarcity of complexes. The dataset was obtained by querying PDB on 2022/06/02 for structures having between 2 and 9 chains with release dates after 2018/04/30. For completeness, we included structures with 2–3 chains as well, although no assembly was performed for these as a minimum of 4 chains is required to create trimeric subcomponents.

The proteins with less than 50 amino acids and structures containing DNA or RNA were excluded. To perform homology reduction between this dataset and the AlphaFold-Multimer[20] training dataset, we used MMseqs2 (release 13-45111)[30] within each oligomeric state with a sequence identity threshold of 30%. If all the chains from a protein structure in this dataset were mapped to a single protein in the training dataset, it was removed. This generated a dataset with 931 dimers, 164 trimers, 269 tetramers, 103 pentamers, 91 hexamers, 74 heptamers, 73 octamers, and 28 nonamers. We then sampled up to 50 complexes from each set randomly to reduce the computational cost (all 28 from the nonamers), resulting in 278 complexes. All MSAs are generated on 8 Intel Xeon E5-2690v4 cores for a maximum 18 h (only a few cases need this long time, most finish within 3-4 h). See above for the folding GPU hardware limitations.

## Human structures in the PDB

To analyse the number of available human PDB files (reported in the introduction), we downloaded all human entries from the PDB on the 14th of October 2021 and counted the number of chains occurring in each entry. In total, there are 2649 human PDB files, 1557 with one chain, 720 with two, and 372 entries with over two chains.

## Hu.MAP

To analyse the gap in complex structural knowledge for human proteins (introduction), all complexes with at least three chains from hu.MAP 2.0[4] were selected. hu.MAP is the result of a machine learning framework that identifies protein complexes using data from over 15,000 mass spectrometry experiments. In total, there are 6956 complexes and 30,572 protein chains, from 9962 unique genes. There are 4779 complexes with at least three chains, of which only 83 have all chains together in the same PDB entry.

## Computational time

The computational time required to predict a complex is mainly limited by the number of subcomponents. For AFM, more MSAs are generated, resulting in ~20× longer runtimes compared to running only HHblits against Uniclust30 (7884 s vs 338 s on average using 16 CPU cores from an Intel Xeon E5-2690v4[43]). The folding takes 1–2 h on NVIDIA A100 Tensor Core GPUs, largely depending on the size of the complexes and MSAs. The complete prediction per subcomponent is thereby in the range of 1–2 h for FoldDock and 3–4 h for AFM. The average number of subcomponents is 13 and 49 using all native and possible trimers for the 175 complexes with 10–30 chains (8561/175 and 2246/175), respectively. This results in a total prediction time of 13–26 or 50–100 and 39–52 or 150–200 h for the guided or all trimeric constituents of each complex with FoldDock and AFM, respectively. The assembly time is identical regardless of how the

subcomponents are predicted, on average 0.34 and 3.73 h per complex using the native trimeric and all trimeric subcomponents, respectively, predicted with AFM on a 2.35 GHz CPU (16 cores of AMD Epyc 7742 CPUs), scaling exponentially as with the number of possible paths. The assembly time is therefore negligible compared to the time needed to predict the structure of the subcomponents and only relies on CPU.

For the complexes with 4–9 chains, the computational time per subcomponent will follow that of the complexes with 10–30 chains. The average assembly time is much less, however, on average 0.12 h per complex. In both cases, the assembly using MCTS is negligible compared to the computation required to predict the subcomponents with FoldDock or AFM. The fast MSA generation using FoldDock, makes this protocol approximately twice as efficient compared to AFM.

## Comparison with other methods

In order to obtain a performance comparison for the presented method, we tested similar existing multi-chain assembly algorithms over the same dataset. We selected Multi-LZerD[27] and Haddock[28] given their high-level performance, ability to deal with large complexes, and availability of code for local installation. We obtained local versions of both Multi-LZerD pipeline (version 2022-06, relying on LZerD version 5.0) and Haddock (version 2.4, relying on CNS version 1.3[44]).

Concerning Haddock, we found that the usage of this tool is oriented toward adopting all available knowledge in the form of restraints. While this procedure may be beneficial to drive docking and reduce computational time, it is impractical to adopt for a large number of structures composing our dataset. To circumvent this limitation, Haddock also allows the possibility to include random surface restraints or centre of mass restraints to force contacts between molecules. As suggested by the authors, we attempted these strategies while increasing the sampling in it0 and it1 stages to 10,000 and 400 models, respectively. For Haddock, 77 complexes were completed.

The Multi-LZerD pipeline required on average 5 h per chain on its preparation stage running standard LZerD, plus several additional hours for the Multi-LZerD stage. This running time was enough to exceed our 72 h maximum allocation time limit, while trying to dock a 10-chain complex, one of the smallest in our dataset. Test runs have been executed over an entire cluster node with 2 Intel Xeon Gold 6130 CPU having 16 cores each and 96GB RAM.

## Data availability

All information needed to repeat the study presented here as well as the pipeline itself is available at: https://gitlab.com/patrickbryant1/molpc. The exact version used for this submission is available at: https://doi.org/10.5281/zenodo.6367019. This repository also contains all figures and the PDB files of the assemblies. All PDB files, MSAs, and pIDDT of the predicted subcomponents and the assemblies for the all-trimer approach are available through this figshare repository: https://doi.org/10.17044/scilifelab.19375172. We cannot provide all data and MSAs for all analyses due to space limitations on all publically available data hosting servers. In total, the data exceeds 10 Tb. The structures corresponding to the PDB-codes mentioned in the main text are available through these links: Figures 1 and 2: 6ESQ (acetoacetyl-CoA thiolase/HMG-CoA synthase complex) Figure 3: 5TRM (Crystal structure of human GCN5 histone acetyltransferase domain), 2V5H (complex of PII and acetylglutamate kinase from Synechococcus elongatus), 7JQZ (Cfl2 wild-type). 5XPB (Selenomethionine labelled Drep4 CIDE domain), 2GRE (Deblocking aminopeptidase), 5T11 (space group C2). Figure 4: 5OVS (BPH), 2X2V (F1Fo-ATP synthase rotor ring), 1DPS (DPS), 1L0L (Mitochondrial cytochrome bc1 complex), 1MFR (M Ferretin), 5XPB (Drep4 CIDE domain). Source data are provided with this paper.

## Code availability

The code and instructions for running MoLPC using the best option with FoldDock are provided in its own github repository to keep the size down: https://github.com/patrickbryant1/MoLPC. There is also a web version provided through a Colab notebook: https://colab.research. google.com/github/patrickbryant1/MoLPC/blob/master/MoLPC.ipynb. The script analyse.sh (https://gitlab.com/patrickbryant1/molpc/-/blob/ main/src/analysis/analyse.sh) reads all data and makes all figures and analyses reported here.

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

## Acknowledgements

Financial support: Swedish Research Council for Natural Science, grant No. VR-2016-06301 and VR-2021-03979, Swedish E-science Research Centre, and the Knut and Alice Wallenberg foundation. Computational resources: Swedish National Infrastructure for Computing, grants: SNIC 2021/5-297, SNIC 2021/6-197, and Berzelius-2021-29. A.E. received all financial support and computational resources.

## Author contributions

P.B. designed and performed the studies. P.B. and A.E. performed the main analysis. G.P., W.Z., A.S., and P.B. set up the modelling infrastructure with AFM. G.P. analysed the performance of LZerd and Haddock. W.Z. and A.S. analysed the performance of AFM for the smaller complexes of 2–9 chains. P.K. assisted in the generation and analysis of the smaller complexes of 2–9 chains. P.B. wrote the first draft of the manuscript and prepared all figures, which were later edited and improved by A.E. and P.B. A.E. obtained funding.

## Funding

## Competing interests

The authors declare no competing interests.
