## [Peer Review File · Nature Communications]

REVIEWERS' COMMENTS

Reviewer #1 (Remarks to the Author):

The authors addressed most of the reviewers comments.

It would still be useful to understand the reasons to the assembly failure. Does it happen mainly because AlphaFold does not produce enough interactions or because they are not picked up by the Monte Carlo Tree Search?

Reviewer #2 (Remarks to the Author):

The authors have addressed my comments, and I have no further ones. The new version is thorough, exhaustive, and acknowledges limitations.

Reviewer #3 (Remarks to the Author):

All my comments have been properly addressed and the manuscript overall has been greatly improved.

REVIEWERS' COMMENTS

Reviewer #1 (Remarks to the Author):

The authors addressed most of the reviewers comments.

It would still be useful to understand the reasons to the assembly failure. Does it happen mainly because AlphaFold does not produce enough interactions or because they are not picked up by the Monte Carlo Tree Search?

We thank the reviewers for their insightful comments, resulting in a significantly improved manuscript.

In the previous revision, we highlighted the relationship between the TM-score of the subcomponents constituting a complex and that of the final assembled complex using complete complexes (Figure 4f). When the subcomponents display high accuracy, so do the assembled complexes (SpearmanR=0.8). To extend this analysis and relate it with complex accuracy using MCTS, we calculate a ROC curve using assemblies with TM-score above 0.8 as positives. The AUC is 0.85 for all complexes and 0.88 for the complete assemblies. We also add an analysis of the incomplete assemblies showing a similar correlation to that of only the complete assemblies (SpearmanR=0.8 for the complete, vs 0.79 for all). Supplementary figure 5 displays these results. Together, this shows that when the accuracy of the subcomponents is high, MCTS can assemble them with high accuracy.

Supplementary Figure 5. a) Average TM-score of subcomponents vs TM-score of the whole complex for all assemblies (n=58) using the native trimers predicted with FoldDock. When the subcomponents display high accuracy, so do the assembled complexes (Spearman R = 0.79). **b)** ROC curve, where positives are assemblies of TM-score ≥ 0.8 , as a function of the average subcomponent TM-score using all and only the complete assemblies, respectively. The AUC is 0.85 for all complexes (positives = 71, negatives = 104) and 0.88 for the complete assemblies (positives = 28, negatives = 30).